# On the Ionic Conductivity of Cation Exchange Membranes in Mixed Sulfates Using the Two-Phase Model

**DOI:** 10.3390/membranes13100811

**Published:** 2023-09-26

**Authors:** Liansheng Wu, Haodong Jiang, Tao Luo, Xinlong Wang

**Affiliations:** Ministry of Education’s Research Center for Comprehensive Utilization and Clean Process Engineering of Phosphorous Resources, School of Chemical Engineering, Sichuan University, Chengdu 610065, China

**Keywords:** ionic conductivity, cation exchange membrane, two-phase model, counter-ion, membrane microstructure

## Abstract

The concentration dependence of the conductivity of ion exchange membranes (IEMs), as well as other transport properties, has been well explained by the contemporary two-phase model (Zabolotsky et al., 1993) considering a gel phase and an inter-gel phase filled with electroneutral solution. Here, this two-phase model has been adopted and first applied in electrolytes containing mixed counter-ions to investigate the correlation between the membrane ionic conductivity and its microstructure. For three representative commercial cation exchange membranes (CEMs), the total membrane conductivity ($\kappa^{T}$) when in equilibrium with mixed MgSO_4_ + Na_2_SO_4_ and H_2_SO_4_ + Na_2_SO_4_ electrolytes could be well predicted with the experimental composition of counter-ions in the gel and inter-gel phase, as well as the counter-ion mobility in the gel phase when the membrane is in a single electrolyte. It is found that the volume fraction of the inter-gel phase ($f_{2}$) has little impact on the predicted results. The accuracy of the model can be largely improved by calculating the inter-gel phase conductivity ($\kappa^{in}$) with the ionic mobility being the same as that in the external solution (obtained via simulation in the OLI Studio), rather than simply as equivalent to the conductivity of the external solution ($\kappa^{s}$). Moreover, a nonlinear correlation between the CEMs’ conductivities and the counter-ion composition in the gel phase is observed in the mixed MgSO_4_ + Na_2_SO_4_ solution, as well as for the Nafion117 membrane in the presence of sulfuric acid. For CEMs in mixed MgSO_4_ + Na_2_SO_4_ electrolytes, the calculated conductivity values considering the interaction parameter σ, similar to the Kohlrausch’s law, are closer to the experimental ones. Overall, this work provides new insights into membrane conductivity with mixed counter-ions and testifies to the applicability of the contemporary two-phase model.

## 1. Introduction

Ion exchange membranes (IEMs) are one of the most advanced membranes and now are widely applied in several separation technologies due to their high permselectivity and energy-efficient performance [1,2,3]. It is known that the IEMs, including the so-called homogeneous ones, are actually spatially nonuniform [4]. This nonuniformity on the sub-microscopical scale will eventually influence the properties of IEMs. Therefore, many researchers in the IEMs field have been exploring the correlation between the fundamental membrane properties and their microstructures by introducing static and dynamic parameters [5,6,7,8].

In early studies, the IEMs are considered as a single homogeneous mixture of matrix polymer chains, ion-exchanged and mobile ions, and water [9,10]. However, the oversimplification of this mathematical model makes it difficult to predict the phenomenological coefficients. Considering the membrane microstructural inhomogeneity, the IEMs could be regarded as a system of two or several microphases [11,12,13]. Then the membrane properties could be correlated with the microphases within the membrane. Ionic conductivity is one of the most essential properties of IEMs [14,15], which has been found as a function of the corresponding microphase properties [16]. According to the different ionic states (ions due to ion exchange with fixed-charge groups and due to the Donnan sorption) within IEMs, A. Yamauchi et al. divided the total conductivity of membranes into two parts: the ion-exchanged conductivity and the Donnan electrolytes conductivity [17]. Based on the Nernst–Planck equation [18], V. I. Zabolotsky et al. [7] developed the contemporary two-phase model for the transport properties, including conductivity, by introducing the structural parameters into the theoretical framework. The model was initially formulated and tested for 1:1 strong electrolytes. N.P. Gnusin et al. [19] successfully applied this contemporary two-phase model in describing the membrane electro-transport properties and in solving the dynamic problems of electrodialysis. Y. Sedkaoui et al. [20] determined the volume fraction of the inter-gel phase of membranes in NaCl and KCl solutions and provided a literature review of these values in NaCl solution. Indeed, there are experimental results reported by V. V. Sarapulova et al. for the membrane conductivity and diffusion permeability treated using the two-phase model for the cases where the solution contains NaCl, CaCl_2,_ or Na_2_SO_4_ [21,22,23]. However, this model has only been verified in IEMs in equilibrium with a single electrolyte. When such a model is applied to weak electrolytes, or to acids and alkalis instead of salts, the researchers might produce results not anticipated by the model since the model does not take into account the additional chemical interactions. Except for the widely accepted two-phase model, there are also papers developing similar multiphase models [24]. T. W. Xu et al. [6] also proposed a simple evaluation method based on conductivity for evaluating the membrane microstructure and transport parameters. However, due to a series of assumptions made in this framework, this method has been less considered. Nevertheless, the membrane behavior in the mixture of these electrolytes was not studied from this theoretical point of view based on the two-phase model.

Recent studies concentrate more on membrane conductivity measurement techniques and the concentration dependence related to the membrane microstructure [20,25,26,27,28,29]. Theoretical considerations about the correlation between the membrane structure and its ionic conductivity are relatively less reported [30]. In this work, the contemporary two-phase model has been adopted and applied to three representative commercial cation exchange membranes (CEMs) in two different mixed electrolytes (MgSO_4_/Na_2_SO_4_ and H_2_SO_4_/Na_2_SO_4_) under a total sulfate concentration range of 0.1~1.0 M. It is known that the proton exhibits an unusual transport mechanism: except for the diffusion, migration, and convection in a typical electro-membrane process [9], there also exists the so-called structural diffusion (Grotthuss) mechanism [31]. Therefore, the contemporary two-phase model is also applied for the mixed electrolytes containing sulfuric acid, which has been rarely reported in literature. Overall, the structural parameters in the contemporary two-phase model will be systematically investigated in the mixed electrolytes, and the correlation between the CEM microstructure and the ionic conductivity will also be discussed here.

## 2. Theoretical Background

### 2.1. Contemporary Two-Phase Model

The importance of studying the effect of IEM structural inhomogeneity has drawn much attention [13]. Based on the thermodynamic knowledge of IEMs [32], the membrane is generally considered as being composed of two microphases, according to the contemporary two-phase model: the gel phase composed of functional charged groups and hydrophilic matrix polymer chain, and the inter-gel phase filled with the Donnan electrolytes [7].

According to the work of V. I. Zabolotsky et al. [7], as shown in Figure 1, the membrane total conductivity ($\kappa^{T}$) is the sum of corresponding conductivities from two parts: the gel phase conductivity ($\kappa^{g}$) and the inter-gel phase conductivity ($\kappa^{in}$). Here, $\kappa^{in}$ is implicitly taken as the conductivity of the external solution ($\kappa^{s}$). Thus, the total conductivity is further expressed by introducing the structural parameters (Equation (1)).
(1)$$\kappa^{T}={\left(f_{1}{{(\kappa }^{g})}^{\alpha }+f_{2}({\kappa^{s})}^{\alpha }\right)}^{1/\alpha }$$
where $f_{1}$, $f_{2}$ represents the volume ratio of the gel and the inter-gel phase, respectively, and $f_{1}+f_{2}=1$. α is the structural parameter, which reflects the reciprocal arrangement of the microphase elements within the membrane. Two extreme conditions, α = 1 and −1, correspond to the situation when all the gel and the inter-gel microphase elements are arranged in parallel and in series, respectively, to the flow direction of the current. Actually, Equation (1) is generalized from these two extreme conditions to reflect the actual arrangement situation of the microphase elements in the membrane.

The previous study, ref. [7] found that the total conductivity depends slightly on α. When $\left|\alpha \right|\ll 1$, Equation (1) becomes:(2a)$$\kappa^{T}={{(\kappa }^{g})}^{f_{1}}{{(\kappa }^{s})}^{f_{2}}$$

Then a linear relationship can be anticipated in a $\ln\kappa^{T}-\ln\kappa^{s}$ correlation:(2b)$$ln\kappa^{T}=f_{1}{ln\kappa }^{g}+f_{2}ln\kappa^{s}$$

However, the correlation above (Equation (1)) has only been verified in a single-electrolyte solution with one monovalent counter-ion. As far as the authors would know, this work is the first to investigate its applicability in a mixed electrolytes system.

In equilibrium with a single electrolyte, the membrane’s gel phase conductivity is dominated by the counter-ion’s concentration and its mobility in the gel phase:(3)$$\kappa_{i}^{g}=z_{i}Fc_{i}^{g}u_{i}^{g}=Fc^{g}u_{i}^{g}$$
where $z_{i}$ is the valence of counter-ion *i*, F is the Faraday constant, $c_{i}^{g}$ is the molarity of counter-ion *i* in the gel phase (mol·m^−3^). $u_{i}^{g}$ is the mobility of counter-ion *i* and $c^{g}$ is the equivalent counter-ion concentration in the gel phase (equ·m^−3^).

In our recent work [33], for mixed electrolytes containing two different counter-ions, the conductivity of the gel phase is determined by the mobilities of the two counter-ions and the interaction between them. Then, two situations exist: 

(I) If the interaction between different counter-ions is negligible, the conductivity of the gel phase is expressed as:(4)$$\kappa_{i,j}^{g}=z_{i}Fc_{i}^{g}u_{i}^{g}+z_{j}Fc_{j}^{g}u_{j}^{g}$$

Then, the total conductivity of the IEM in equilibrium with mixed electrolytes could be expressed by combining Equations (2a) and (4).
(5)$$\kappa^{T}={\left(z_{i}Fc_{i}^{g}u_{i}^{g}+z_{j}Fc_{j}^{g}u_{j}^{g}\right)}^{f_{1}}{\left(\kappa^{s}\right)}^{f_{2}}$$

Here, the mobility of counter-ion *i* ($u_{i}^{g}$) is obtained through Equation (3) when the IEM is in a single electrolyte. The concentration of counter-ion ($c_{i}^{g}$) is determined experimentally.

(II) If the interaction between counter-ions is non negligible, the Kohlrausch’s law is then employed for the description of the counter-ion mobility variations [34].
(6)$$\kappa_{i,j}^{g}=x_{i}^{m}\left(\kappa_{i}^{g}-\sigma \sqrt{x_{j}^{m}}\right)+x_{j}^{m}\kappa_{j}^{g}$$

σ is the interaction parameter, which is determined by the experimental conductivity data. $x_{i}^{m}$ is equivalent molar ratio of counter-ion *i* in the gel phase and is expressed as $x_{i}^{m}=z_{i}c_{i}^{g}/(z_{i}c_{i}^{g}+z_{j}c_{j}^{g})$. Then, Equation (2a) is further expressed as:(7)$$\kappa^{T}={\left(x_{i}^{m}\left(\kappa_{i}^{g}-\sigma \sqrt{x_{j}^{m}}\right)+x_{j}^{m}\kappa_{j}^{g}\right)}^{f_{1}}{\left(\kappa^{s}\right)}^{f_{2}}$$

### 2.2. The Inter-Gel Phase Conductivity

In Section 2.1, the conductivity of the inter-gel phase is simply considered equivalent to the conductivity of the external solution, which is implicitly indicated in the work of V.I. Zabalotsky et al. [7]. However, the value of the inter-gel phase conductivity has a direct influence on the structural parameter fitting results in the contemporary two-phase model. Here, the inter-gel phase conductivity is also calculated via two different approaches.

In the first approach, the ionic mobilities in the infinitely diluted solution are applied in calculating the inter-gel phase conductivity ($\kappa^{in}$):(8)$$\kappa^{in}=F\sum \left(z_{+}c_{+}^{D}u_{+}^{dilute}+z_{-}c_{-}^{D}u_{-}^{dilute}\right)$$
where $c_{+}^{D}$, $c_{-}^{D}$ are the concentrations of cations and anions, respectively, due to the Donnan sorption. $u_{+}^{dilute}$, $u_{-}^{dilute}$ are the mobility of corresponding ions in the infinitely diluted solution. Table 1 lists the values of ionic mobilities employed in this work [3].

In the second approach, the conductivity of the inter-gel phase is also calculated by assuming that the mobilities of ions comprising the Donnan electrolytes are the same as the corresponding ions’ mobilities in the external solution. In a single-electrolyte solution, its ionic conductivity is expressed as:(9)$$\kappa^{s}=F(z_{+}c_{+}^{s}u_{+}^{s}+z_{-}c_{-}^{s}u_{-}^{s})$$

$c_{+}^{s}$, $c_{-}^{s}$ are the concentrations of cations and anions in the external solution; $u_{+}^{s}$, $u_{-}^{s}$ are the ionic mobilities in the external solution, respectively. According to the electro-neutrality of the external solution, $z_{+}c_{+}^{s}=z_{-}c_{-}^{s}=c_{equ}^{s}$, Equation (9) becomes:(10)$$\kappa^{s}=Fc_{equ}^{s}(u_{+}^{s}+u_{-}^{s})$$
where $c_{equ}^{s}$ is the equivalent electrolyte concentration in the external solution. Similarly, the conductivity of the inter-gel phase is expressed as:(11)$$\kappa^{in}=Fc_{equ}^{D}(u_{+}^{D}+u_{-}^{D})$$
where $c_{equ}^{D}$ is the equivalent Donnan electrolyte concentration in the inter-gel phase. $u_{+}^{D}$, $u_{-}^{D}$ are the ionic mobilities within the Donnan electrolytes, respectively.

Considering that the ionic mobilities are consistent in both the Donnan electrolytes and the external solution ($u_{+}^{s}=u_{+}^{D}$, $u_{-}^{s}=u_{-}^{D}$), the value of the inter-gel phase conductivity could be calculated from the conductivity of the external solution, as:(12)$$\kappa^{in}=\kappa^{s}\frac{c_{equ}^{D}}{c_{equ}^{s}}$$

Then the total conductivity of IEMs could be predicted by the fitting results of the $\kappa^{T}-\kappa^{in}$ correlation as well as by replacing the $\kappa^{s}$ in Equations (5) and (7) with $\kappa^{in}$.

## 3. Materials and Methods

### 3.1. Cation Exchange Membranes

Three representative commercial CEMs with different microstructures are investigated in this work. The CSE membrane (Astom Co., Tokyo, Japan) is a standard homogeneous CEM used for desalination. Nafion117 is a representative perfluorosulfonic acid membrane from Chemours, America. A heterogeneous membrane, 3361BW (Shanghai Shanghua Water Treatment Material Co., Shanghai, China), is also investigated. The fundamental properties of the membranes have been determined and reported in our previous report [33], and are also provided here in Table 2. 

### 3.2. Membrane Ionic Composition

The composition of both the Donnan electrolytes within the inter-gel phase and counter-ions in the gel phase has been investigated under a total sulfate concentration range of 0.1~1.0 M. Figure 2 schematically shows the experimental procedures for determining the CEM ionic composition. Before the partitioning experiments, all the membrane samples are cut into 4 × 4 cm and immersed in 0.5 M Na_2_SO_4_ solution for 48 h to convert the membrane into Na^+^ form. Then, all the samples are washed several times with deionized water and transferred to the mixed-electrolyte solutions. After equilibrium in a mixed-electrolytes solution for 48 h, the membrane surface moisture is quickly removed with filter papers. Next, the membranes are immersed in excessive deionized water for 24 h to completely elute the Donnan electrolytes within the inter-gel phase, and all the elution is collected to give a final volume of 250 mL. Then, the wet membranes are prepared for another step: the determination of the counter-ions’ composition in the gel phase. For the mixed MgSO_4_/Na_2_SO_4_ electrolytes system, a membrane sample is ion-exchanged with 100 mL 0.5 M MgSO_4_ solution twice, while the membrane sample in the mixed H_2_SO_4_/Na_2_SO_4_ electrolytes system is ion-exchanged with 100 mL 0.5 M H_2_SO_4_ twice. All the elution solutions are then gathered to give a total volume of 250 mL. The concentration of cations in the solution is analyzed by ICP-OES (inductively coupled plasma, optically emission spectrometer, PE Avio 200, Waltham, MA, USA) or an automatic potentiometric titrator (916, Metrohm, Herisau, Switzerland).

### 3.3. Conductivity Measurement

Based on electrochemical impedance spectroscopy (EIS), a direct-contact method is applied to determine the through-plane ionic conductivity of CEMs. The conductivity cell installation (Figure 3) is the same as that used in our previous work [33]. After equilibrium in the external solution, the investigated membrane is removed from the solution and directly sandwiched between two electrodes. Then, the EIS spectra are acquired from an electrochemical workstation (Zennium IM6, Zahner, Gundelsdorf, Germany), together with a buffer unit supplying an AC current of 5 mV amplitude in the frequency range of 1~300 kHz. All the measurements are conducted at room temperature (25 °C). More details can be found in our previous report [33]. 

## 4. Results and Discussions

### 4.1. Determination of the Membrane Structural Parameters

Before applying the contemporary two-phase model in mixed electrolytes, the membrane structural parameters, $f_{2}$ and α, fitted from either the nonlinear (Equation (1)) or the linear (Equation (2b)) correlation, need to be determined. The total conductivity ($\kappa^{T}$) of CEMs in a single electrolyte is measured under a sulfate concentration range of 0.1 ~ 1.0 M. The actual conductivity of the external solution ($\kappa^{s}$) is obtained from the OLI Studio simulation results. In this work, the membrane structural parameters α and the volume fraction of two membrane microphases $f_{i}$ are obtained through two different approaches.

#### 4.1.1. Fitted from the External Solution Conductivity $\kappa^{s}$

As illustrated in Figure 4a, a good linear relationship can be observed in the $\ln\kappa^{T}-\ln\kappa^{s}$ correlation for the heterogeneous membrane 3361BW under different single-electrolyte solutions. The slop represents the volume fraction of the inter-gel phase ($f_{2}$). It is clear that $f_{2}$ changes with the type of the external electrolyte. The value of $f_{2}$ for the 3361BW membrane reaches its maximum in the H_2_SO_4_ solution (~0.48). However, for the homogeneous CEMs (CSE and Nafion117, Figure 4b,c), generally poor linear fitting results are observed.

Considering that the linear relationship is not so obvious for all three investigated CEMs under different electrolytes, the nonlinear correlation method (Equation (1)) has also been applied in acquiring the membrane structural parameters. Table 3 shows the nonlinear fitting results for the CEMs under different electrolytes. Though the variance of the nonlinear fitting is overall larger than the linear fitting results, the value of $f_{2}$ is too large to be real for some CEMs under a certain electrolyte. For example, for the heterogenous membrane 3361BW in sulfuric acid, $f_{2}=$ 0.818, for the CSE membrane under MgSO_4_, $f_{2}=$ 0.998, and under H_2_SO_4_, $f_{2}=$ 0.999. These unrealistic values of $f_{2}$ would be very likely numerical artifacts of the nonlinear fitting. In the report of V. I. Zabolotsky et al. [7], the value of α is in the range of 0.1~0.3 for most CEMs. It is observed here that the fitting values of α for CEMs could be negative, especially in sulfuric acid solution. These negative values can be anticipated and have been reported in the work of N.D. Pismenskaya et al. [30], and they are discussed in the review of V. Nikonenko et al. [35]. 

#### 4.1.2. Fitted from the Inter-Gel Phase Conductivity $\kappa^{in}$

In Section 4.1.1, the conductivity of the inter-gel phase ($\kappa^{in}$) is simply taken as equivalent to the external solution conductivity $\kappa^{s}$. According to our previous research [33], the composition of the Donnan electrolytes within the membrane is in fact not the same as the external solution. Therefore, in order to better describe the relationship between the CEM conductivity and the membrane microstructure, $\kappa^{in}$ is also acquired by assuming that the ionic mobility of the Donnan electrolytes is the same as that in the external electrolyte. 

The membrane structural parameter results calculated via two different approaches (Equations (8) and (12)) are shown in Table 4 and Table 5, respectively. Different from the results fitted with the external solution conductivity $\kappa^{s}$ (Table 3), the $f_{2}$ value here is in the range of 0 ~ 0.3, which is consistent with the literature results [4,7,36]. Interestingly, the values of $\left|\alpha \right|$ for CEMs in H_2_SO_4_ electrolyte solution here are all greater than 1, which is not reported in previous research. Numerical artifact alone would not explain these unphysical values that contradict the model. At the moment, we do not have a clear understanding of why $\left|\alpha \right|$ would be larger than 1. Moreover, there are some cases in which the nonlinear fitting process does not converge (the missing data in Table 5). Therefore, the membrane structural parameters fitted from the linear method are applied in predicting the total conductivity of CEMs under mixed electrolytes in subsequent sections. 

### 4.2. Prediction of Membrane Conductivity in Mixed Electrolytes

#### 4.2.1. $\kappa^{T}$ Prediction with the External Solution Conductivity $\kappa^{S}$

In Section 4.1, the membrane structural parameters fitted via different approaches with the membrane conductivity data in the single-electrolyte solution have been systematically discussed. Then the conductivity of CEMs could be predicted through Equation (5) or Equation (7).

Figure 5 and Figure 6 show the variation of the total membrane conductivities with the cation composition of the external mixed-sulfate solutions. Here, the total membrane conductivities are obtained both experimentally (via the direct-contact method) and theoretically (Equation (5)). It is indicated from the results that the calculated values provide a relatively accurate prediction of the trend in the total conductivity of CEMs under mixed electrolytes (the red solid line in the graph represents the trend of the predicted values). Interestingly, the predicted values are overall larger than the experimental ones, whether in the mixed MgSO_4_ + Na_2_SO_4_ or the H_2_SO_4_ + Na_2_SO_4_ electrolytes.

As can be seen in Section 4.1 (Table 4 and Table 5), the value of $f_{2}$ will change with the type of external electrolyte. Therefore, the fitted values of $f_{2}$ from different electrolytes are adopted here in predicting the total conductivities of CEMs under mixed electrolytes. The orange, blue, and red symbols in Figure 5 and Figure 6 represent the total membrane conductivities calculated with different $f_{2}$ values. It is clear that the variation of $f_{2}$ has little impact on the predicted membrane ionic conductivity, especially for the perfluorosulfonic acid membrane Nafion117. Overall, the difference between the predicted and experimental values is relevant to the membrane’s microscopic channels and the water content. For the heterogeneous membrane 3361BW, with a larger inter-gel phase (Table 4 and Table 5) and higher water content (Table 2), the accuracy of the contemporary two-phase model is obviously lower than for the homogeneous CEMs.

#### 4.2.2. $\kappa^{T}$ Prediction with the Inter-Gel Phase Conductivity $\kappa^{in}$

In Section 4.2.1, the predicted total conductivities of CEMs are obtained by simply incorporating the conductivity of the external solution ($\kappa^{s}$) into the contemporary two-phase model. Here, the membrane total conductivity is also predicted by assuming that the ionic mobility in the inter-gel phase is the same as that in the external solution. 

As mentioned in Section 4.1.2, there are two different approaches to calculating the ionic mobility in the inter-gel phase. In the first approach, the ionic mobility under infinite dilution is used for calculating the $\kappa^{in}$ (Equation (8)). As shown in Figure 7a, the results predicted via this approach with the contemporary two-phase model are contrary to the experimental ones for the CSE membrane under mixed MgSO_4_ + Na_2_SO_4_ solutions. This discrepancy can be attributed to the over-simplification of the ionic mobility in the inter-gel phase and that under infinite dilution (Table 1). The results for other CEMs (3361BW and Nafion117) are provided in the Appendix A.

In the second approach, the mobilities of ions in the inter-gel phase are represented with the actual mobilities of ions in the corresponding external solutions (Equations (10)–(12)). Then, $\kappa^{in}$ is obtained from the simulated conductivity data of the external solution ($\kappa^{s}$) analyzed in the OLI Studio, together with the experimental composition of the Donnan electrolyte within the membrane ($c_{equ}^{D}$, Equation (12)). It is obvious from the revised results (Figure 7b) that the accuracy of the contemporary two-phase model can be largely improved by employing the actual ionic mobility in the external solution, especially for the CSE membrane in the di-/monovalent cations (Mg^2+^/Na^+^) system. Thus, the second approach is reasonable and suggested.

Furthermore, the numerical results of the cations mobility in the gel phase calculated from two different approaches (Table 6) can better explain the discrepancy observed in Figure 7a. First, the fitting results of the $\ln\kappa^{T}$ − $\ln\kappa^{in}$ correlation (Equation (2b)) for a single-electrolyte solution give the conductivity of the membrane gel phase ($\kappa^{g}$). Then, values of the cation mobility in the membrane gel phase ($u_{i}^{g}$) can be calculated via Equation (3). The two indications in Table 6, “infinite dilution” and “OLI simulation”, represent the approaches when $\kappa^{in}$ is calculated with the ionic mobility in the inter-gel phase as the value under infinite dilution and as the value the same as that in the external solution simulated in the OLI Studio. It can be concluded that the main reason for the conductivity discrepancy of the CSE membrane is the different Mg^2+^ mobility in the membrane phase. 

Figure 8 and Figure 9 show the total conductivity results of the CEMs under two different kinds of mixed electrolytes. Compared to the method of simply incorporating the solution conductivity ($\kappa^{s}$) into the contemporary two-phase model (Figure 5 and Figure 6), the results obtained with the ionic mobility in the inter-gel phase as identical to that in the external solution are closer to the experimental values, especially for the mixed-electrolyte solutions containing sulfuric acid. However, for the heterogenous membrane 3361BW under the mixed MgSO_4_ + Na_2_SO_4_ electrolytes, the accuracy of this approach is lower than that for the homogenous CEMs.

The numerical data of predicted CEMs’ total conductivity in mixed-sulfate solutions of different total sulfate concentration are provided in the Appendix A. 

### 4.3. Correlation between Conductivity and Counter-Ions Composition in the Gel Phase

In the discussions above, the interaction between different counter-ions in the membrane gel phase is ignored. However, a nonlinear correlation between the conductivity and the counter-ions’ composition in the gel phase of IEMs has been proved both in our previous work [33,37] and in the literature [34,38]. In this work, Kohlrausch’s law (Equation (6)) is introduced to describe this nonlinear relationship. It is shown in Figure 10 that the nonlinearity between the membrane gel phase conductivity and the cation composition is non-negligible for three CEMs under mixed MgSO_4_ + Na_2_SO_4_ electrolytes, especially when the total sulfate concentration is low (0.1 M). 

Interestingly, a perfect linear relationship is observed for the 3361BW and CSE membranes in mixed electrolytes containing sulfuric acid (Figure 11a,b). The gel phase conductivity hardly changes with the total sulfate concentration in the external solution. However, for Nafion117 (Figure 11c), nonlinear correlations can clearly be observed for different total sulfate concentrations, which is consistent with the results in the literature [33,34]. The numerical fitting results are provided in the Appendix A.

Therefore, the contemporary two-phase model revised with Kohlrausch`s interaction parameter σ (Equations (6) and (7)) has also been applied in calculating the CEMs’ total conductivities under mixed electrolytes of different compositions. Figure 12 shows the calculated total membrane conductivities. Compared with the linear method (Equation (5) and Figure 8), the total membrane conductivity results calculated with the $\kappa^{g}$ values obtained from this nonlinear correlation are closer to experimental ones for CEMs under mixed MgSO_4_ + Na_2_SO_4_ electrolytes. The total conductivity values of CEMs in mixed-electrolyte solutions containing sulfuric acid are provided in the Appendix A. It is observed from these results that the interaction of counter-ions in the membrane gel phase for the Na^+^/Mg^2+^ pair is more obvious than that for the Na^+^/H^+^ pair.

## 5. Conclusions

In this work, the contemporary two-phase model of ion exchange membranes (IEMs) has been systematically investigated for three representative cation exchange membranes (CEMs) in equilibrium with mixed-sulfate electrolytes of two different counter-ions (Na^+^/Mg^2+^ and Na^+^/H^+^) using a total sulfate concentration of 0.1~1.0 M. After experimentally obtaining the membrane ionic composition, the total conductivity of CEMs ($\kappa^{T}$) in mixed electrolytes could be well predicted via the two-phase model. Though the membrane structural parameter f in this model changes with the nature of the external electrolyte, the volume fraction of the inter-gel phase $f_{2}$ has little impact on the calculated membrane total conductivity. The accuracy of the two-phase model could be largely improved by replacing the external solution conductivity ($\kappa^{s}$) with the inter-gel phase conductivity ($\kappa^{in}$). A further improvement can be achieved by calculating the $\kappa^{in}$ as the Donnan electrolytes’ conductivity, with the ionic mobility in the inter-gel phase being the same as that in the external solution, rather than simply with the ionic mobilities under infinite dilution, especially for the CSE membrane under mixed MgSO_4_ + Na_2_SO_4_ electrolytes. Moreover, the calculated total conductivity revised with the interaction parameter (σ) derived from Kohlrausch’s law is closer to the experimental values in mixed MgSO_4_ + Na_2_SO_4_ electrolytes, while the linear correlation method is more suitable for mixed-sulfate electrolytes in the presence of sulfuric acid. Overall, the contemporary two-phase model could be successfully applied in mixed-sulfate electrolytes for CEMs, and the accuracy of this model is better for mixed mono-/monovalent counter-ion pairs than for di-/monovalent counter-ions pairs.

## Figures and Tables

**Figure 1 membranes-13-00811-f001:**
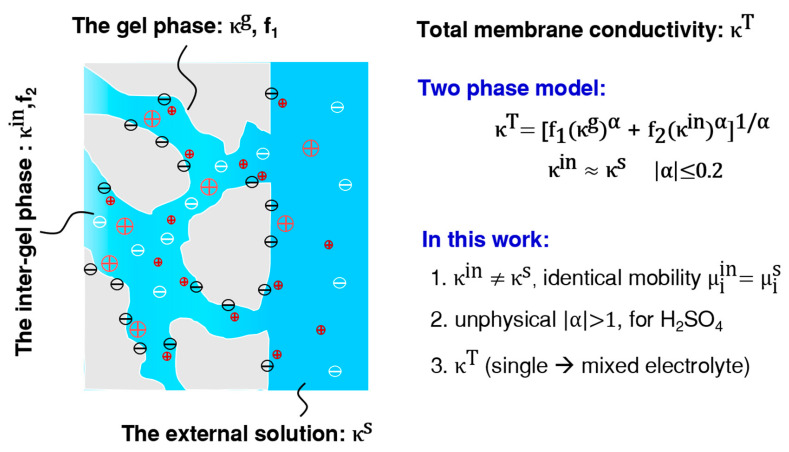
The correlation between the total ionic conductivity of cation exchange membranes (CEMs), $\kappa^{T}$, and the gel phase ($\kappa^{g}$) and the inter-gel phase conductivity ($\kappa^{in}$) according to the contemporary two-phase model. New attempts are also summarized here. $\left|\alpha \right|$ > 1 results from nonlinear fitting according to Equation (1) and it contradicts the model.

**Figure 2 membranes-13-00811-f002:**
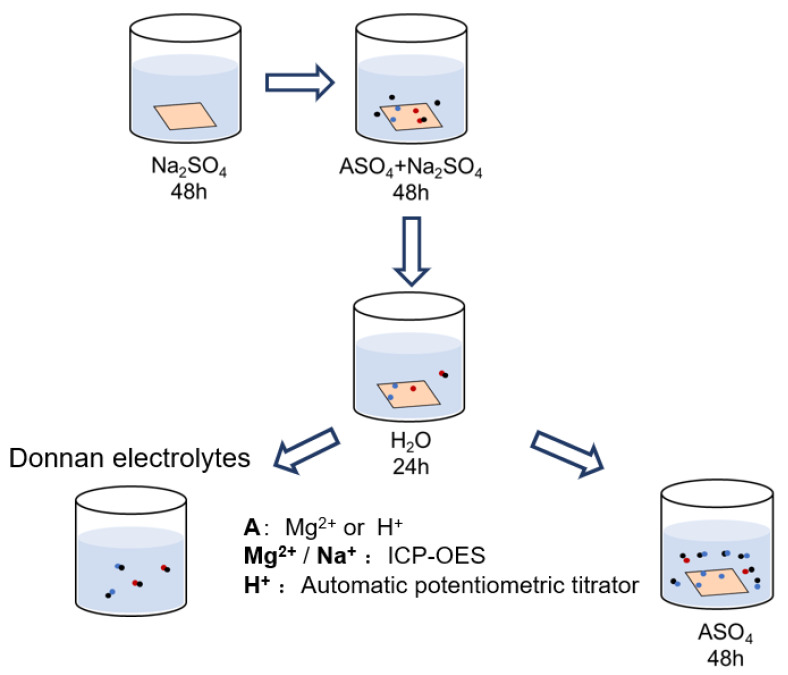
Schematical illustration of the experimental procedures for determining the ionic composition of CEMs after equilibrium in mixed-sulfate solutions.

**Figure 3 membranes-13-00811-f003:**
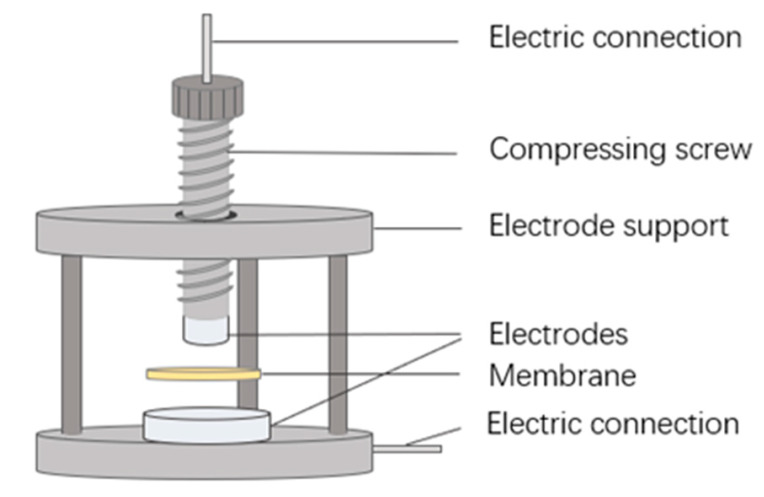
Simple illustration of the cell configuration for the measurement of the through-plane membrane resistance (reproduced with permission from Ref. [33], Elsevier 2023).

**Figure 4 membranes-13-00811-f004:**
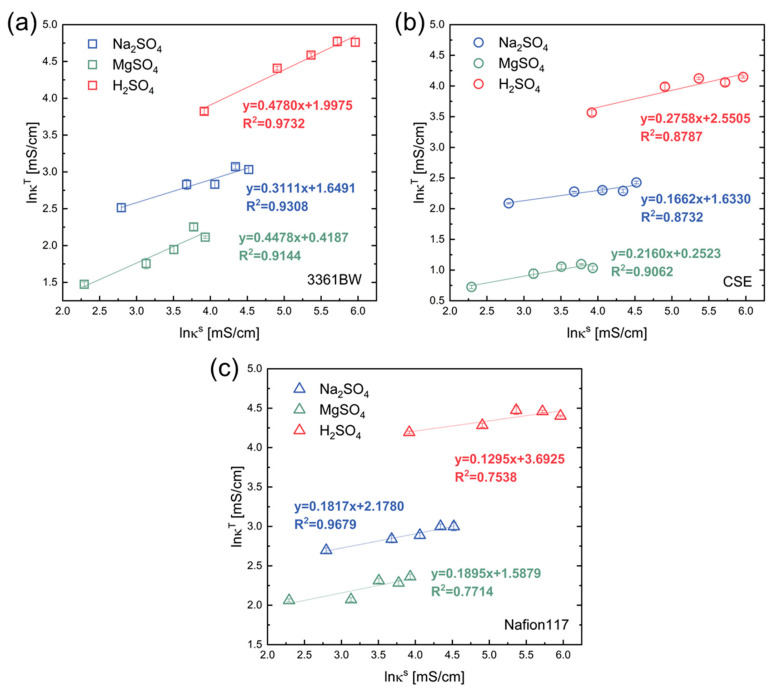
The linear correlation between the ionic conductivities of three CEMs, (**a**) 3361BW, (**b**) CSE and (**c**) Nafion117, and the conductivity of an external solution with a single sulfate electrolyte in the $\ln\kappa^{T}-\ln\kappa^{s}$ coordinates.

**Figure 5 membranes-13-00811-f005:**
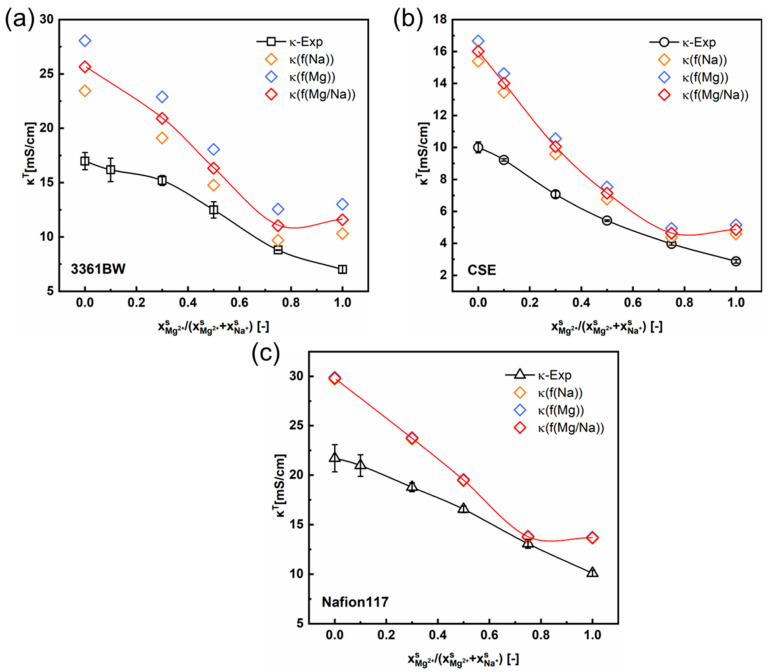
The variation of the experimental and the predicted total membrane conductivities with the cation composition of the external mixed MgSO_4_ + Na_2_SO_4_ electrolytes of 0.5 M total sulfate, for the 3361BW membrane (**a**), CSE membrane (**b**) and Nafion 117 membrane (**c**). Here, *f*(Na) (or *f*(Mg)) represents the volume fraction of the inter-gel phase in the membrane fitted from the single Na_2_SO_4_ (or MgSO_4_) electrolyte solution, and *f*(Mg/Na) is the arithmetic average value.

**Figure 6 membranes-13-00811-f006:**
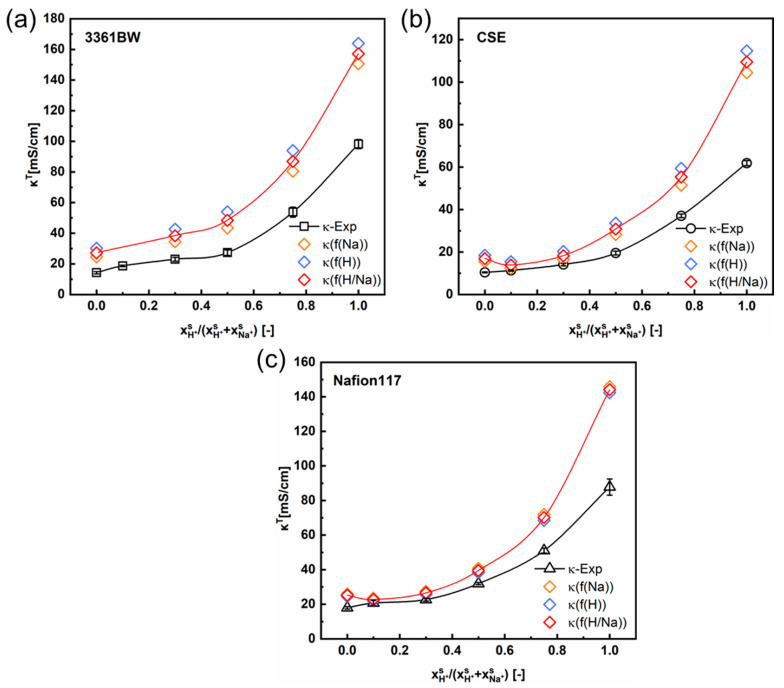
The variation of the experimental and the predicted total membrane conductivities with the cation composition of the external mixed H_2_SO_4_ + Na_2_SO_4_ electrolytes of 0.5 M total sulfate, for the 3361BW membrane (**a**), CSE membrane (**b**) and Nafion 117 membrane (**c**). Here, *f*(Na) (or *f*(H)) represents the volume fraction of the inter-gel phase in the membrane fitted from the single Na_2_SO_4_ (or H_2_SO_4_) electrolyte solution, and *f*(H/Na) is the arithmetic average value.

**Figure 7 membranes-13-00811-f007:**
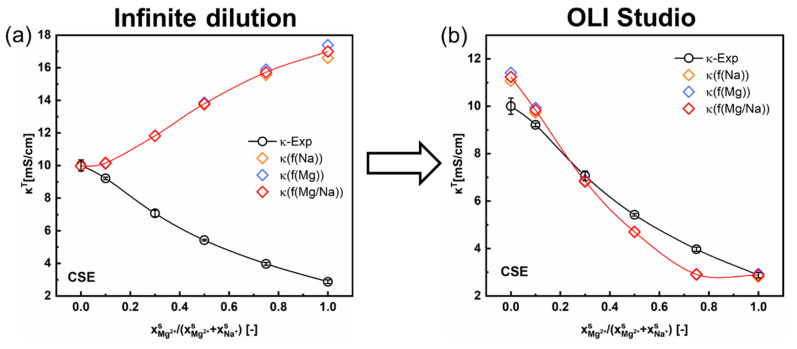
Comparison of the total conductivity variations for the CSE membrane in mixed MgSO_4_ + Na_2_SO_4_ electrolytes as a function of the cation composition in the external solution. The inter-gel phase conductivity ($\kappa^{in}$) is obtained via two different approaches: calculated with (**a**) the ionic mobilities in infinite dilution, (**b**) with the inter-gel phase conductivity data simulated in the OLI Studio.

**Figure 8 membranes-13-00811-f008:**
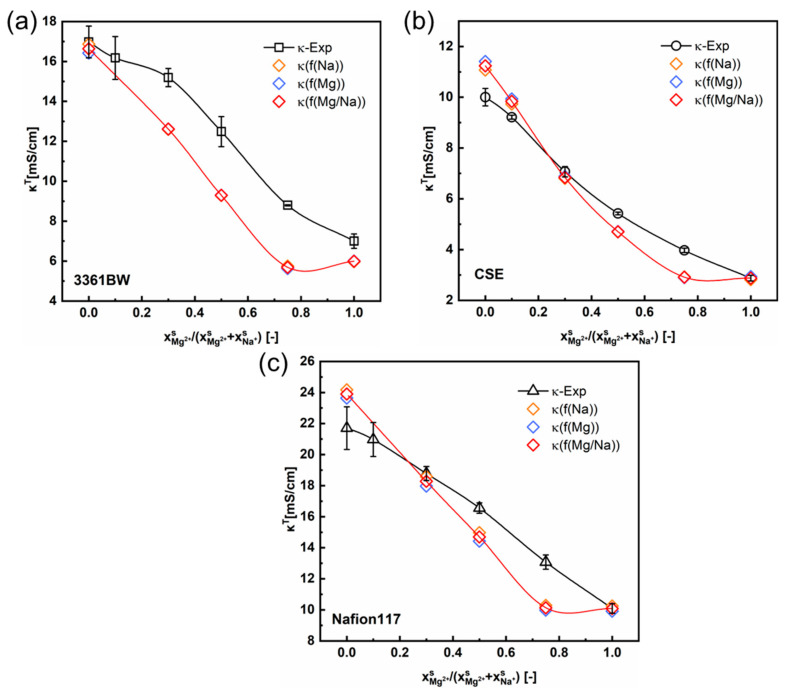
The variation of the experimental and the predicted total conductivities of (**a**) 3361BW, (**b**) CSE and (**c**) Nafion117 with the cation composition of the external mixed MgSO_4_ + Na_2_SO_4_ electrolytes of 0.5 M total sulfate. The ionic mobilities in the inter-gel phase are the same as those in the external solutions obtained from the OLI Studio data.

**Figure 9 membranes-13-00811-f009:**
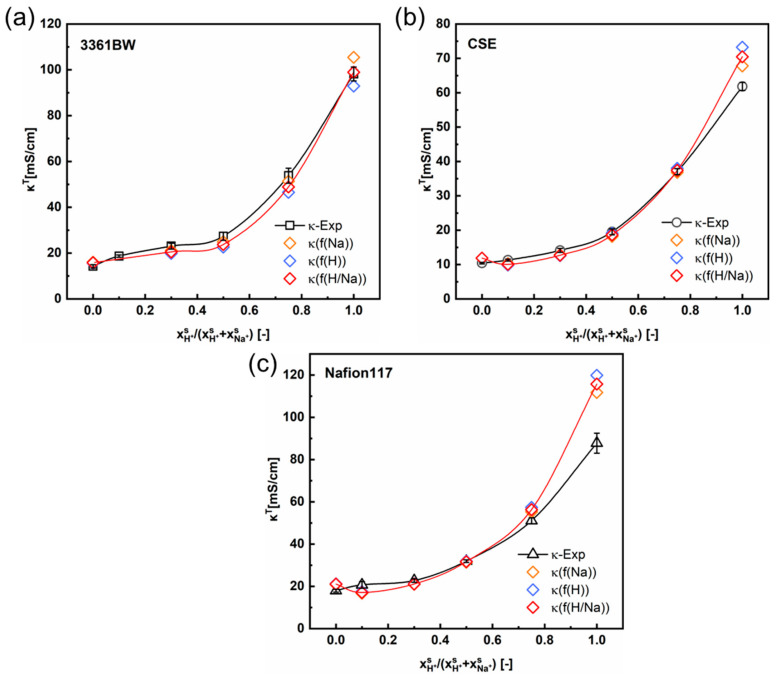
The variation of the experimental and the predicted total conductivities of (**a**) 3361BW, (**b**) CSE and (**c**) Nafion117 with the cation composition of the external mixed H_2_SO_4_ + Na_2_SO_4_ electrolytes of 0.5 M total sulfate. The ionic mobilities in the inter-gel phase are the same as those in the external solutions obtained from the OLI Studio data.

**Figure 10 membranes-13-00811-f010:**
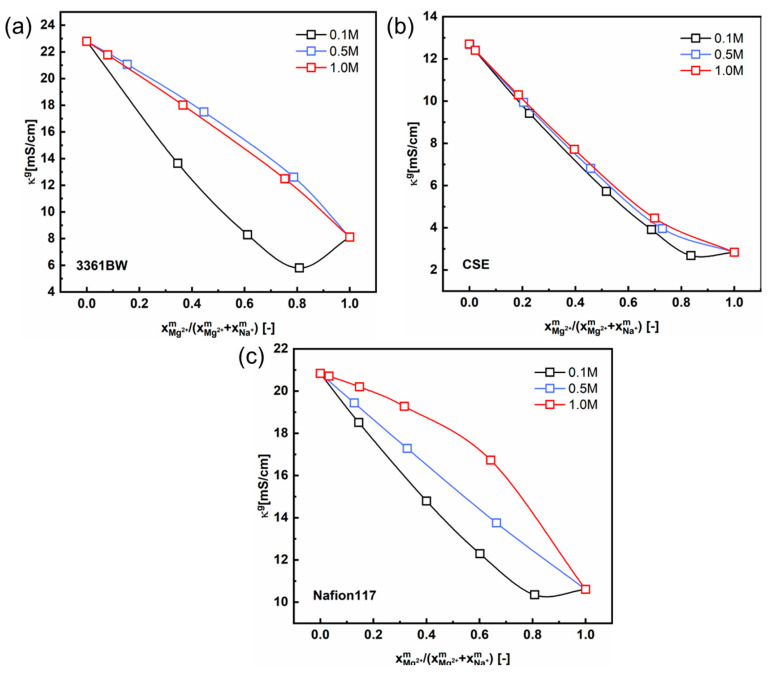
The correlation between the gel phase conductivity ($\kappa^{g}$) of CEMs, (**a**) 3361BW, (**b**) CSE and (**c**) Nafion117, and the counter-ions composition in the gel phase under mixed MgSO_4_ + Na_2_SO_4_ electrolytes of different total sulfate concentrations (0.1, 0.5 and 1.0 M).

**Figure 11 membranes-13-00811-f011:**
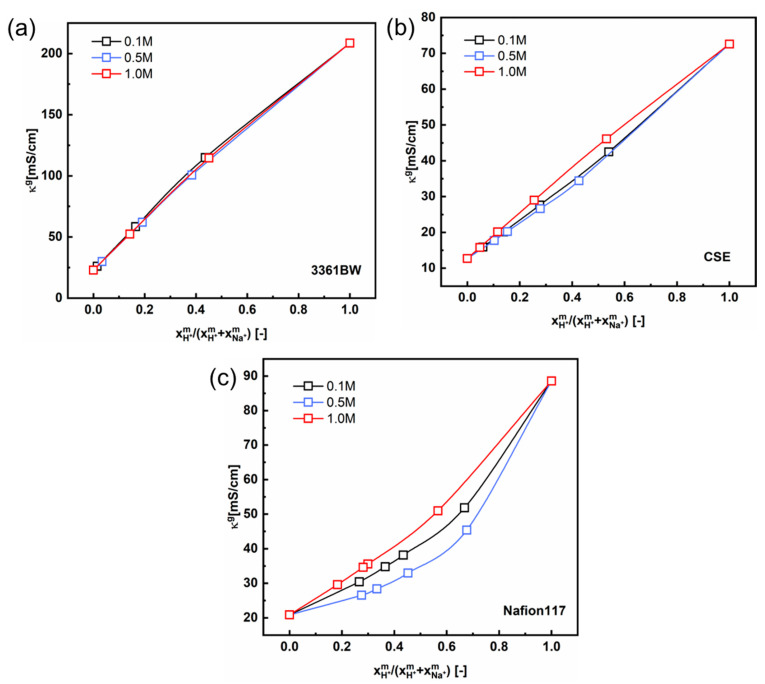
The correlation between the gel phase conductivity ($\kappa^{g}$) of CEMs, (**a**) 3361BW, (**b**) CSE and (**c**) Nafion117, and the counter-ions composition in the gel phase under mixed H_2_SO_4_ + Na_2_SO_4_ electrolytes of different total sulfate concentrations (0.1, 0.5 and 1.0 M).

**Figure 12 membranes-13-00811-f012:**
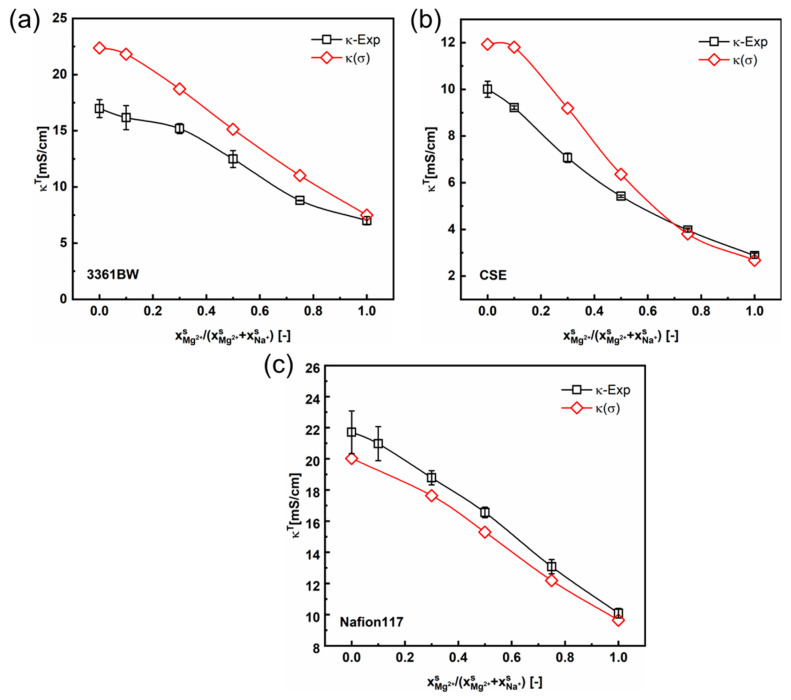
The variation of the experimental and the predicted total membrane conductivities with the cation composition of the mixed external MgSO_4_ + Na_2_SO_4_ electrolytes (total sulfate concentration 0.5 M), for the 3361BW membrane (**a**), CSE membrane (**b**) and Nafion 117 membrane (**c**). The gel phase conductivity ($\kappa^{g}$) of the CEMs is obtained from the nonlinear correlation method considering the interaction parameter (α).

**Table 1 membranes-13-00811-t001:** Ionic mobilities in infinite dilution at 298.15 K.

Ions	Na^+^	Mg^2+^	H^+^	SO_4_^2−^
$u_{i}\times 10^{8}$ (m^2^s^−1^V^−1^)	5.20	5.50	36.2	8.27

**Table 2 membranes-13-00811-t002:** Fundamental properties of commercial cation exchange membranes.

Mem.	Area Resistance (Ω cm^2^)	Density of Dry Polymer (g cm^−3^)	Dry Membrane Thickness (μm)	Water Uptake (g g^−1^ Dry Polymer)	IEC (meq g^−1^ Dry Polymer)
CSE	1.8	1.16 ± 0.03	160	0.29 ± 0.05	2.34 ± 0.01
3361BW	11.0	0.93 ± 0.01	420	0.43 ± 0.08	2.45 ± 0.01
Nafion117	1.0	1.94 ± 0.11	180	0.19 ± 0.01	0.99 ± 0.01

**Table 3 membranes-13-00811-t003:** The fitting results of the nonlinear correlation between $\kappa^{T}$ and $\kappa^{s}$ according to Equation (1).

Membranes	Electrolytes	α [-]	$f_{2}$ [-]	R^2^ [-]
3361BW	Na_2_SO_4_	0.025	0.305	0.910
	MgSO_4_	0.281	0.317	0.868
	H_2_SO_4_	−1.072	** *0.818* **	0.987
CSE	Na_2_SO_4_	−0.006	0.168	0.852
	MgSO_4_	−0.732	** *0.998* **	0.924
	H_2_SO_4_	−1.561	** *0.999* **	0.959
Nafion117	Na_2_SO_4_	0.427	0.128	0.971
	MgSO_4_	1.180	0.058	0.833
	H_2_SO_4_	−0.734	0.188	0.750

**Table 4 membranes-13-00811-t004:** The fitting results of the nonlinear and linear correlations between $\kappa^{T}$ and $\kappa^{in}$, and the $\kappa^{in}$ is calculated from the ionic mobility under infinite dilution (Equation (8)).

Membranes	Electrolytes	Nonlinear			Linear	
		α [-]	$f_{2}$ [-]	R^2^ [-]	$f_{2}$ [-]	R^2^ [-]
3361BW	Na_2_SO_4_	−0.055	0.207	0.916	0.210	0.931
	MgSO_4_	0.213	0.167	0.740	0.157	0.801
	H_2_SO_4_	−1.393	0.155	0.999	0.397	0.932
CSE	Na_2_SO_4_	−0.282	0.141	0.868	0.153	0.863
	MgSO_4_	−0.382	0.099	0.868	0.098	0.879
	H_2_SO_4_	−2.439	0.005	0.884	0.150	0.610
Nafion117	Na_2_SO_4_	0.256	0.120	0.935	0.098	0.931
	MgSO_4_	0.940	0.086	0.785	0.125	0.695
	H_2_SO_4_	−1.257	0.014	0.772	0.068	0.581

**Table 5 membranes-13-00811-t005:** The fitting results of the nonlinear and linear correlations between $\kappa^{T}$ and $\kappa^{in}$, and the $\kappa^{in}$ is obtained from the ionic mobility the same as the external solution (Equation (12)).

Membranes	Electrolytes	Nonlinear			Linear	
		α [-]	$f_{2}$ [-]	R^2^ [-]	$f_{2}$ [-]	R^2^ [-]
3361BW	Na_2_SO_4_	0.162	0.310	0.919	0.269	0.932
	MgSO_4_	0.400	0.319	0.711	0.178	0.769
	H_2_SO_4_	-	-	-	0.440	0.924
CSE	Na_2_SO_4_	−0.356	0.152	0.866	0.218	0.858
	MgSO_4_	−0.379	0.074	0.853	0.118	0.876
	H_2_SO_4_	−1.703	0.006	0.795	0.159	0.588
Nafion117	Na_2_SO_4_	0.477	0.239	0.924	0.117	0.908
	MgSO_4_	-	-	-	0.163	0.640
	H_2_SO_4_	−1.866	0.002	0.754	0.072	0.569

**Table 6 membranes-13-00811-t006:** Mobilities of cations in the gel phase obtained with two different sets of fitting results.

Mem.	$u_{{Na}^{+}}^{g}$× 10^5^ (cm^2^ s^−1^V^−1^)		$u_{{Mg}^{2+}}^{g}$× 10^5^ (cm^2^ s^−1^V^−1^)		$u_{H^{+}}^{g}$× 10^5^ (cm^2^ s^−1^V^−1^)	
	Infinite Dilution	OLI Simulation	Infinite Dilution	OLI Simulation	Infinite Dilution	OLI Simulation
CSE	4.93	5.14	9.32	1.40	33.37	31.23
3361BW	10.85	5.65	0.96	0.41	96.04	34.60
Nafion117	13.24	13.52	1.55	1.95	59.00	57.30

## Data Availability

Data will be available upon request.

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
