# Peer review of "On the Ionic Conductivity of Cation Exchange Membranes in Mixed Sulfates Using the Two-Phase Model"

_membranes, 2023, doi:10.3390/membranes13100811_

Round 1

Reviewer 1 Report

Electrodialysis treatment of complex solutions requires the methods to predict the membrane properties. The problem is that the models are usually formulated and tested for 1:1 strong electrolyte. See for example the article by Szymczyk et al 10.1016/j.memsci.2016.02.003 , which both determines the volume fraction of intergel phase of membranes in NaCl and KCl solutions and provides the literature review of these values in NaCl solution. And when such methods are applied to non-1:1 salts, or to weak electrolytes, or to acids and alkalis instead of salts, the researchers might produce the results not anticipated by the model since the model does not take into account the additional chemical interactions.

It was interesting to review f2 reaching 0.999. Zabolotsky once criticized the presentation with f2 = 0.4 as unrealistically high and said that such membrane would have poor counterion selectivity. I might gather that the unrealistic f2 is an artifact of nonlinear fitting, but the authors might want to elaborate further.

Speaking of fitting, when discussing the square R it might be useful to provide the confidence intervals in electrical conductivity plots, where possible. It might be that the approximation curve fits very nicely in the confidence margins.

When discussing with alpha parameter I must agree with the authors, and alpha values I saw mostly lay in range 0.1-0.3. However, the authors of the present manuscript accentuate that in some instances they obtained the negative alpha values. I should point out that possibility of negative alpha values was anticipated by creators of microheterogeneous model; see for example doi:10.3390/app9010025 : " α is the structural parameter depending on the position of the phases with respect to the axis of transport: when the phases are parallel to this axis, α = 1; when they are in serial disposition, α = −1; in other cases −1 < α < 1.". Similar phrasing is also in doi:10.1134/S0965544118060087 , and it seems that such phrasing was also used in earlier works not translated in English. The physics behind the range is explained by the form of equation (1) in the present manuscript and it becomes visible when alpha = 1 and alpha = -1 is substituted in the equation. So while the negative values of alpha do fit the model, the absolute values of alpha exceeding 1 are interesting. Can the authors please elaborate on the physical meaning of such values?

I am also interested in phrase "Interestingly, the values of |α| for CEMs in H2SO4 electrolyte solution here are all greater than 1, which is rarely reported in previous researches". May the authors please provide references that do report such high absolute values?

---

I thank the authors for presented interesting experimental results and for the work done on their model interpretation. Best regards.

Reviewer 2 Report

See attachment 

Round 2

Reviewer 2 Report

All questions are answered and incorporated in the text.